# Information content in time series of litter decomposition studies and the transit time of litter in aridlands

Agustín Sarquis[1, 2] and Carlos A. Sierra[3]

[1]Facultad de Agronomía, Universidad de Buenos Aires, Buenos Aires, 1417, Argentina
[2]Instituto de Investigaciones Fisiológicas y Ecológicas Vinculadas a la Agricultura (IFEVA; CONICET-FAUBA), Buenos Aires, 1417, Argentina
[3]Max-Planck-Institut für Biogeochemie, Jena, 07745, Germany

*Correspondence to:* Carlos A. Sierra (csierra@bgc-jena.mpg.de)

**Abstract.** Plant litter decomposition stands at the intersection between carbon (C) loss and sequestration in
terrestrial ecosystems. During this process organic matter experiences chemical and physical transformations that affect decomposition rates of distinct components with different transformation fates. However, most decomposition studies only fit one-pool models that consider organic matter in litter as a single homogenous pool and do not incorporate the dynamics of litter transformations and transfers in their framework. As an alternative, compartmental dynamical systems are sets of differential equations that serve to represent both
the heterogeneity in decomposition rates of organic matter, and the transformations it can undergo. This is achieved by including parameters for initial proportion of mass in each compartment, their respective decomposition rates, and mass transfer coefficients between compartments. The number of compartments, as well as their interactions, in turn, determine model structure. For instance, a one-pool model can be considered a compartmental model with only one compartment. Models with two or more parameters, in turn, can have
different structures, such as parallel if each compartment decomposes independently, or in series if there is mass transfer from one compartment to another. However, because of these differences in model parameters, comparisons in model performance can be complicated. In this context we introduce the concept of transit time, a random variable defined as the age distribution of particles when they are released from a system which can be used to compare models with different structures. In this study, we first asked what model structures are
more appropriate to represent decomposition from a publicly available database of decomposition studies in aridlands: *aridec*. For this purpose, we fit one- and two-pool decomposition models with parallel and series structures, compared their performance using the Bias Corrected Akaike Information Criteria (AICc), and used model averaging as a multi-model inference approach. We then asked what the potential ranges of the median transit times of litter mass in aridlands are and what are their relationships with environmental variables. Hence,
we calculated median transit time for those models and explored patterns in the data with respect to mean annual temperature and precipitation, solar radiation, and the Global Aridity Index. Median transit time was 1.9 years for the one- and two-pool model with parallel structure, and five years for the two-pool series model. The information in our datasets supported all three models in a relatively similar way, thus our decision to use a multi-model inference approach. After model-averaging, median transit time had values of around three years
for all datasets. Exploring patterns of transit time in relation to environmental variables yielded weak correlation coefficients, except for mean annual temperature, which was moderate and negative. Overall, our analysis suggests that current and historical litter decomposition studies often do not contain information on how litter quality changes over time or do not last long enough for litter to entirely decompose. This makes fitting accurate mechanistic models very difficult. Nevertheless, the multi-model inference framework proposed here can help
to reconcile theoretical expectations with the information content from field studies and can further help to design field experiments that better represent the complexity of the litter decomposition process.

## 1 Introduction

Plant litter decomposition is the process through which plant-derived organic matter is broken down into smaller components. The main biotic driver of decomposition is the metabolic activity of fungi and bacteria (Bradford et
al., 2017), but soil fauna can be important too (García-Palacios et al., 2013; Zanne et al., 2022). The magnitude of biotic decomposition is further determined by climate (Gholz et al., 2000) and litter quality (Cornwell et al., 2008). Additionally, abiotic drivers of decomposition like solar radiation can have a large contribution to this process (Méndez et al., 2022). Altogether, plant litter decomposition releases carbon that was fixed by plants back to the atmosphere and mediates soil organic matter formation (Cotrufo et al., 2015). This puts

decomposition at a crucial intersection between C loss and sequestration in terrestrial ecosystems. It is thus of great interest to gain a better understanding on how decomposition influences the terrestrial C balance and how this process would be affected by global change.

Plant litter is composed of material of different physical and chemical properties that decays at different rates (Adair et al., 2008; Tuomi et al., 2009). However, litter decomposition models commonly assume a single pool

that considers the decomposition of organic matter as if it was a homogenous mass pool with a single decomposition constant (Adair et al., 2010). Alternatively, organic matter dynamics can be modelled using compartmental dynamical systems, which are sets of differential equations that serve to represent both the heterogeneity of organic matter chemical quality, and the transformations plant residues can undergo (Sierra and Müller, 2015). This is achieved with the inclusion of different pools that decompose at different rates. This

allows to model the dynamics of labile C compounds that are more readily available for microbial consumption like sugars, and other compounds that have a longer persistence in the litter pool like tannins or lignin. Additionally, it is possible to include interactions between these pools like C transfers from one pool to another. This mass transfer between pools represents the transformation of molecules in litter without actual mass loss from the litter system (Prescott and Vesterdal, 2021). The number of compartments, as well as their interactions,

finally determine model structure. Compartmental models of decomposition have been successfully applied for decades (Chappelle et al., 2023; Parton et al., 1987; Tuomi et al., 2009), and it has been proven many times that they can be an improvement from the traditional one-pool model (Adair et al., 2008; Cornwell and Weedon, 2014; Derrien and Amelung, 2011; Manzoni et al., 2012).

Despite the richness of information that can be learned from compartmental models, there are still limitations

for their widespread application. One main limitation is parameter identifiability. This happens because more complex models usually have more parameters and, in some cases, the information contained in time series of litter mass loss may not be enough to estimate those parameters unambiguously (Brun et al., 2001). Depending on the resolution and extension of the time series, it might be possible to obtain different number of parameters from the available data (Sarquis et al., 2022a; Sierra et al., 2015). Consequently, different studies developed

under different methodologies and sampling schemes may provide information on different model structures. Further, this limits the application of compartmental models to data from extensive heterogenous databases, since not all parameters might be identifiable for all datasets (Sarquis et al., 2022a).

It is common to compare model parameters like the decomposition constant when the same model has been applied to many datasets. But, comparing the behavior of models with different structures in the same way is

not possible, because decomposition constants of single homogenous pools are not comparable to decomposition constants of specific pools, such as those in compartmental models. Thus, a metric that can be used to compare models with different structures is the transit time of mass in a complex heterogeneous system. Transit time represents the mean age of particles when they are released from a system (Sierra et al., 2017). In the context of litter decomposition studies, transit time can tell us about how long it takes for mass to exit litter

since the start of an experiment. Transit time is a random variable with its own probability distribution, and thus mean and median transit times can be calculated (Sierra et al., 2018). Unlike a single decomposition rate, transit time can be calculated for the bulk of litter when using compartmental models. Transit time contains information from all different mass compartments (Lu et al., 2018), and so, it becomes a more useful parameter when making comparisons from models that have different structures.

In this study we used the *aridec* database, which is an open access database of published decomposition studies in aridlands from around the world (Sarquis et al., 2022a). The focus of this database on aridlands stems from how widespread aridlands are, since around 41% of the land surface is classified as arid to some extent (Safriel and Adeel, 2005). This large area represents a wide range of diverse ecosystems, with many shared functional characteristics. For instance, aridlands are usually more sparsely vegetated (Guttal and Jayaprakash, 2007) and

this produces a shift in the importance of decomposition drivers in comparison to humid ecosystems. Plant litter under these conditions is more susceptible to solar radiation (Austin and Vivanco, 2006) and desiccation by wind (D'Odorico et al., 2019). Further, water sources other than rain can become more relevant when mean annual precipitation is low (Evans et al., 2020). These unique traits of arid ecosystems probably explain why decomposition rates are not correlated to mean annual precipitation in these systems (Austin, 2011), contrary

to what was proposed in the traditional literature (e.g., Meentemeyer, 1978). Furthermore, aridland processes are thought to become more widespread in the future because of aridland expansion (Feng and Fu, 2013) and drought-intensification of humid ecosystems (Grünzweig et al., 2022).

Hence, we used the *aridec* database to address the following questions: given the information content in time series of litter decomposition studies, what model structures are more appropriate to represent decomposition

from arid ecosystems? From the set of models obtained, what are the potential ranges of the median transit times of litter mass? Moreover, what are the potential relationships between median transit time and

environmental variables? We fit one- and two-pool decomposition models with parallel and series structures, compared their performance using AICc, and used model averaging as a multi-model inference approach. We further calculated transit times for those models and explored patterns in the data in relation to environmental variables.

## 2 Methods

### 2.1 Model fitting

First, we used the *aridec* database to fit a group of candidate decomposition models. The *aridec* database is a publicly available database of decomposition studies from aridlands across the world (Sarquis et al., 2022a). This database contains bulk litter mass loss data, but it lacks mass loss dynamics of different litter organic matter pools that decompose at different rates (e.g.: soluble carbohydrates, cellulose, lignin). Because of this, we took an inverse-modelling approach that allowed us to estimate the parameters of these unknown pools by fitting the models to mass loss data. This model calibration procedure constitutes a non-linear optimization problem, where the objective is to find parameter values that minimize a measure of badness of fit, like a weighted sum of squared residuals (Soetaert and Petzoldt, 2010). Following this procedure, we obtained a group of parameters for each dataset and fit the dynamics of mass loss for different pools. We did this with the SoilR (Sierra et al., 2012) and the FME (Soetaert and Petzoldt, 2010) packages in R (R Core Team, 2020).

SoilR is a modelling framework that contains a wide set of functions and tools to model soil organic matter decomposition within the R computing platform. Organic matter decomposition in SoilR is represented by systems of linear differential equations that generalize most compartment-based models. A simple general structure to represent litter decay with no inputs follows Equation 1:

$$\frac{dC(t)}{dt} = A\,C(t) \tag{1}$$

$$C(t) = [C_{pool\,1}, \,...,\, C_{pool\,m}]^{\mathrm{T}}$$

$$A = \begin{bmatrix} -k_1 & \cdots & a_{1i} \\ \vdots & \ddots & \vdots \\ a_{j1} & \cdots & -k_m \end{bmatrix}$$

Where $C(t)$ is a $m \times 1$ vector with $m$ pools of litter mass observed at time $t$, and $A$ is a square $m \times m$ matrix that contains decomposition rates ($k_m$) for each pool and transfer rates ($a_{ij}$) between them. These different pools may correspond to different ways in which the quality of the litter is expressed in different studies. For example, they may correspond to different compounds obtained from a specific extraction method (e.g.: water soluble sugars, or acid detergent lignin), or they can be defined by general decay classes such as fast and slow decay compounds. The linear dynamical system represented by Eq. (1), has many different solutions, but we are only interested in the solution that satisfies

$$C\,(t = 0) = C_0 = [total\ C_0 \cdot p_1, \,...,\, total\ C_0 \cdot p_m]^T \tag{2}$$

where $C_0$ is a $m{\times}1$ vector with the value of initial litter mass content in the different compartments $m$. We set total initial $C_0$ to be 100% for this analysis and the resulting $p_m$ parameters are the initial proportions of litter in $m$ pools.

Before fitting the models, we run a collinearity test following the procedure by Soetaert and Petdzolt (2010) and the results are presented in Sarquis et al. (2022a). Briefly, when parameters are functionally related, changes in one of them can be compensated by changes in others. This produces different parameter sets that have similar probability distributions, thus it is impossible to determine a single parameter set for a model (Brun et al., 2001; Sierra et al., 2015). From this analysis, we were able to choose three models: a one-pool model, a two-pool parallel model, and a two-pool series model (Fig. 1). The one-pool model represents mass loss data as a single homogeneous mass compartment and has a single parameter, the decomposition rate $k$. The two-pool model with parallel structure considers litter mass as two distinct compartments that decompose at different rates. Hence, its parameters are the two decompositions rates ($k_1$ and $k_2$) and the initial proportion of litter mass in pool one ($p_1$, from which the proportion of mass in pool 2 can be calculated as $p_2 = 1 - p_1$). Finally, the two-pool series model is similar to the parallel model, but it incorporates the transfer of matter from pool one to pool

two after its transformation. This is indicated in the model by the parameter $a_{12}$ (i.e., the transfer rate from pool 1 to pool 2).

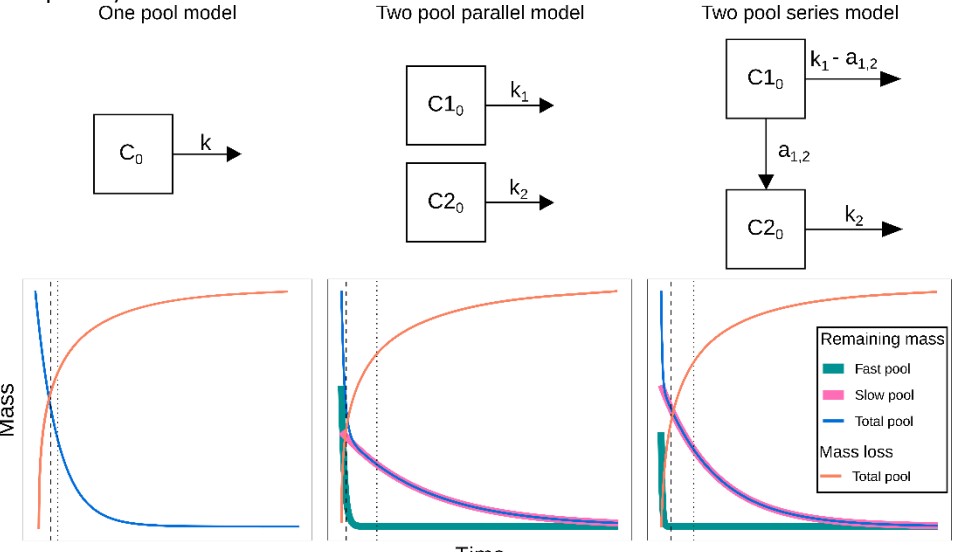

Figure 1: Decomposition models fitted in this study. $C_0$: total initial litter mass; $C1_0$: initial litter mass in the fast-decomposing pool; $C2_0$: initial litter mass in the slow-decomposing pool; $k$, $k_1$, $k_2$: decomposition rates of the total, fast- and slow-decomposing litter pools, respectively; $a_{1,2}$: mass transfer coefficient from the fast-decomposing pool to the slow-decomposing pool; dashed lines denote median transit time; dotted lines denote mean transit time.

Specifically for the two-pool series model our collinearity analysis showed that only 20.1% of the datasets produced identifiable results for this model, and only so when we restricted parameter $p_1$. Restricting or fixing parameters to known values is a way of avoiding collinearity issues. For this purpose, we decided to use initial litter lignin content as a proxy for the $p_2$ parameter (the initial proportion of mass in pool 2) which is complementary to $p_1$ ($p_1 + p_2 = 1$). We were limited by the number of datasets that provided initial lignin values in *aridec*. We searched for this missing information in the TRY database, which contains plant trait data for ecology and earth system sciences (Kattge et al., 2020). We could only find information for three of these datasets in the TRY database. We then completed some of the missing values by averaging lignin data of the same litter types that were already present in *aridec*. Having all the data ready, we proceeded to fit the models mentioned above. All time variables were transformed to monthly timescales to achieve more consistent comparisons.

## 2.2 Transit time

For each model, we calculated litter mass transit time (Sierra et al., 2017). This concept represents the mean age of the particles when they are released from the bulk litter. Another way to interpret this is the time it takes particles to transit the litter system since the beginning of the experiment. We used a modified version of the Mean Transit Time (MTT) from Sierra et al. (2017) without new litter inputs:

$$MTT = -(1, \ldots, 1)\ A^{-1} \tag{3}$$

For both two-pool models, we used the function transitTime in the SoilR package. This function calculates the mean and median of the distribution of the transit time as well as other quantiles of the distribution. The transit time median is interpreted as the time it takes half the litter mass in a sample to decompose. As a special case, for the one-pool model the MTT can be simply calculated as:

$$MTT = \frac{1}{k} \tag{4}$$

While the Median TT (mTT) can be calculated as:

$$mTT = \frac{\ln 2}{k} \tag{5}$$

We found that MTT was usually overestimated in our models (S4, Pre-averaging results table), possibly due to the already slow decomposition rates of arid lands and the inclusion of the $a_{12}$ parameter that prolonged the time that molecules remained in the litter system in the two-pool series models. Instead, values of mTT were

usually lower, so we decided to only work with mTT hereafter. However, some of the mTT values obtained were also overestimated and so we decided to make a cutoff at a mTT of 14.5 years. This value came from fitting the two-pool series model to the longest data set in aridec which is 10 years long and corresponds to average data of different species at Central Plains Experimental Range in Adair et al. (2017) (S1). We excluded from this study the data sets that exceeded this median transit time cutoff. Finally, after accounting for collinearity, the availability of initial litter lignin data and the mTT cutoff, we were left with 128 data sets from 12 aridec entries (Appendix A).

## 2.3 Model selection and multi-model inference

As a first attempt at model selection, we calculated the Bias Corrected Akaike Information Criteria, which is used for small sample sizes (AICc; Burnham and Anderson, 2002). We used the formula from Shumway and Stoffer (2017):

$$AICc = \log \sigma 2k + \frac{n+k}{n-k-2} \tag{6}$$

where $\sigma 2k$ is the variance of the model (in this case the mean squared residuals, i.e. sum of squared residuals divided by sample size), $k$ is the number of parameters in the model, and $n$ is the sample size or the number of points in each time series. We accounted for the variance as one of the parameters in the formula as Burnham and Anderson (2002) recommend.

A common way of choosing the model with the best fit is by looking at the model with the lowest AIC value. We did this by using the *akaike.weights* function from the *qpcR* package. Additionally, we calculated the difference in AICc between the model with the lowest AICc and the other two candidate models (ΔAICc). Since we did not have enough information to choose a single model structure based on AICc (see Results section), we decided to follow a multi-model inference approach (Burnham and Anderson, 2002). We first calculated Akaike Weights using the function *Weights* from the MuMIn R package for each model. Akaike Weights can be interpreted as the probability that a model *j* is the best of all *i* candidate models given the data (Lukacs et al., 2010), and are calculated as:

$$w_j = \frac{\exp\left(-\frac{1}{2}\Delta AICc_j\right)}{\sum_{i=1} \exp\left(-\frac{1}{2}\Delta AICc_i\right)} \tag{7}$$

We then calculated new average estimators for the mean and the median transit times as:

$$\hat{\bar{\beta}}_i = \sum_{j=1} w_j \hat{\beta}_{ij} \tag{8}$$

where $\hat{\beta}_{ij}$ is the $i$ parameter estimator $\hat{\beta}$ for each $j$ model. This results in estimators of mean (*avgMTT*) and median (*avgmTT*) transit times averaged across models for each database entry.

We calculated the unconditional variance as well for each averaged estimator (Burnham and Anderson, 2002; Lukacs et al., 2010) as:

$$v\hat{a}r\left[\hat{\bar{\beta}}_i\right] = \sum_{j=1} w_j \left[MSR_j + \left(\hat{\beta}_{ij} - \hat{\bar{\beta}}_i\right)^2\right] \tag{9}$$

Finally, we estimated 95% confidence intervals as:

$$\hat{\bar{\beta}}_i \pm cv \sqrt{v\hat{a}r\left[\hat{\bar{\beta}}_i\right]} \tag{10}$$

where *cv* stands for the critical value of a t-distribution for a particular number of degrees of freedom.

We made non-parametric Kendall's rank correlation tests between study duration in days and avgMTT and avgmTT, respectively. We also plotted data against environmental variables to explore potential relationships between avgmTT and calculated Pearson *r* correlation coefficients. We used data already available in *aridec* like mean annual temperature and mean annual precipitation. We additionally used Global Aridity Index as calculated in Sarquis et al. (2022a) for aridec entries and annual downward shortwave radiation (hereafter annual solar radiation) from the TerraClimate database (Abatzoglou et al., 2018). We only used data from litter decomposed in ambient conditions (without manipulative treatments) for data exploration.

Further, to test whether the data fit an exponential distribution, we calculated the ratio between *avgmTT* and *ln2 * avgMTT*. In an exponential distribution, the median equals *ln2* times the mean. So, if the ratio between the median from our models (avgmTT) and the median calculated as *ln2 * avgMTT* equals 1, that would imply that

both medians are equal, and the model follows an exponential distribution. All calculations, modelling and figures were made using R (R Core Team, 2020).

**3 Results**

We fit three different candidate models for 128 time series of decomposition, which totaled 384 models. The information in our datasets supported all three models in a similar way. Most times the one-pool model had the lowest AICc values, but close to one-third of the times the two-pool series model had the best fit according to AICc (Fig. 2A and Table S2). Our ΔAICc values were very low (ΔAICc of the 3rd quartile: 1.515), so we would have not been able to apply a ΔAICc=2 cutoff criterion if we wanted to, even when this practice is not recommended (Anderson, 2008; Burnham and Anderson, 2002). All of this showed that the information available was not enough to choose a single model with the best fit. Additionally, we obtained root mean squared residuals for all 128 datasets. For the one-pool model this indicator ranged from 1.1 to 12.9, for the two-pool parallel model it ranged from 1.1 to 12.3, and for the two-pool series model it ranged from 0.3 to 6.6 (Fig. 2B). The first two models performed similarly according to this parameter, but the series model had considerably lower residuals. Following this, we decided to implement a multi-model inference approach using model averaging, which left us with 128 individual models (see Table S3 for model variance and confidence intervals).

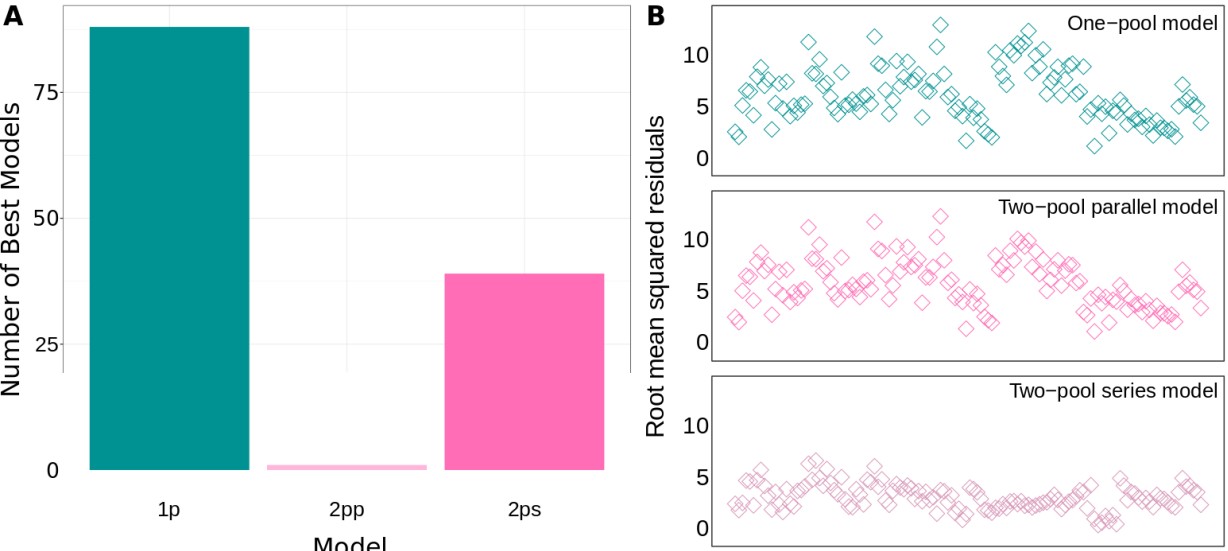

**Figure 2: Model fit for all 128 datasets. (A) Number of models with the lowest AICc values, and (B) root mean squared residuals for each model structure. 1p: one-pool model; 2pp: two-pool parallel model; 2ps: two-pool series model.**

Median transit time of plant litter in arid lands after model averaging was within the range of the original models (Fig. 3). In this analysis, we only used data from litter decomposed in ambient conditions (without manipulative treatments). One and two-pool parallel models had similar mTT (23.27 ± 9.28 and 23.04 ± 9.65 months, mean ± standard deviation respectively). The two-pool series model had near three-fold mTT values of 60.21 ± 45.80 months. After model-averaging mTT (i.e.: avgmTT) dropped to 36.15 ± 22.20 months.

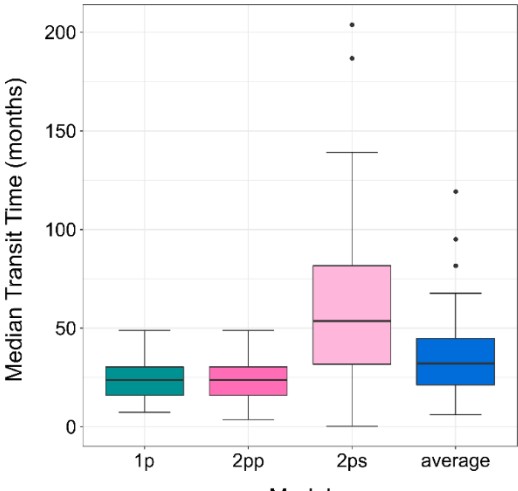

**Figure 3: Median transit time (months) for three different models and for the averaged model. Only data for control or ambient treatments were used for this figure. 1p: one-pool model; 2pp: two-pool parallel model; 2ps: two-pool series model.**

Looking at the avgmTT alone showed the wide range of time that litter takes to decompose in arid ecosystems (Fig. 4). Correlation between duration in days and the avgMTT was positive (tau=0.2, p=0.002) but it was not significantly different from zero for avgmTT (p=0.3; Appendix B). Exploration of patterns of transit time in relation to environmental variables yielded weak correlation coefficients, except for mean annual temperature which was moderate but significative (r = -0.56, p = 0.047). Values of avgmTT at the coldest end ranged between 37 and 65 months, while the warmest site showed values of 8 months (Fig. 4a). This shows that plant litter in warmer aridlands decomposes faster than in colder sites.

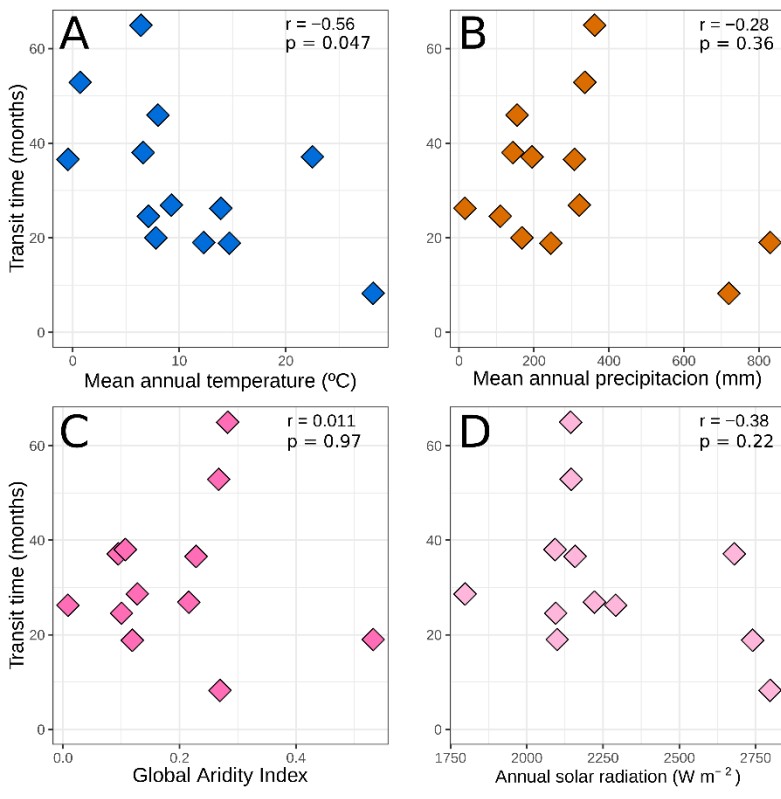

**Figure 4: Transit time (months) Pearson correlations with (A) mean annual temperatures (°C), (B) mean annual precipitation (mm), (C) global aridity index, and (D) annual solar radiation (W m⁻²). Only data for control or ambient treatments were used for this figure. Each diamond represents the mean avgmTT for each site. Pearson correlation coefficients (*r*) and p-values are displayed.**

Calculating the quotient between the avgmTT and avgMTT times the natural logarithm of two showed contrasting results (Fig. 5). Forty-two percent of the models in this analysis had values near to zero, which suggests that those models did not follow an exponential distribution. This is because in an exponential distribution the median equals *ln2* times the mean, and their ratio, if equal, should result in one. On the other hand, only 15 % of the models had values between 0.9 and 1.0. Complementarily, this suggests that those models

had indeed a near exponential distribution.

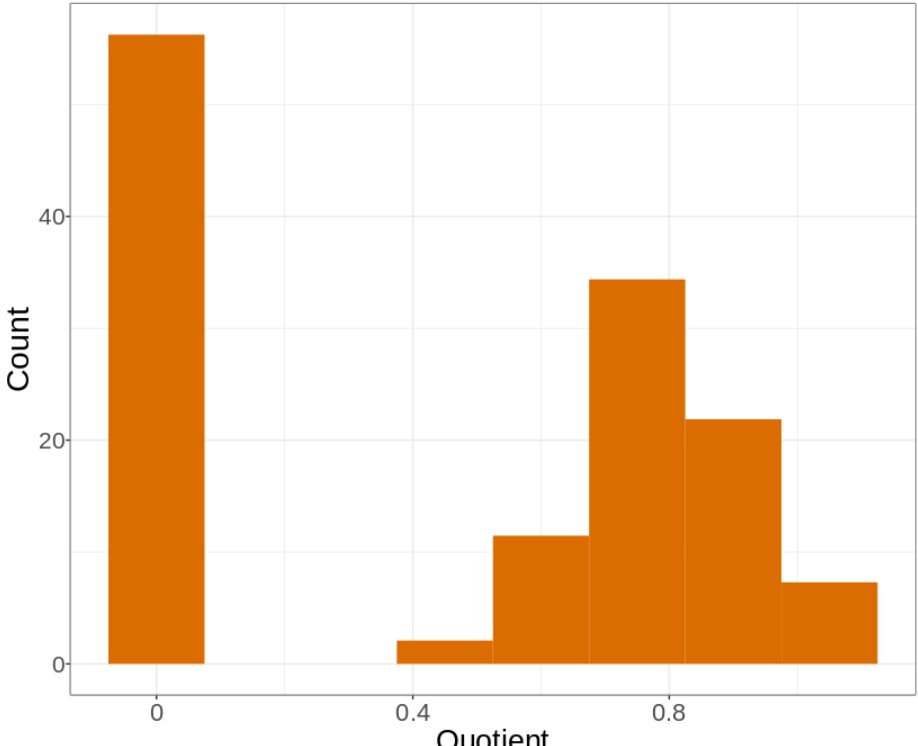

**Figure 5: Histogram of frequency for the quotient of the median transit time and the natural logarithm of two times the mean transit time from average models. Bars represent the number of models for a range of values of the quotient.**

**4 Discussion**

We asked as our first question: what model structures are more appropriate to represent decomposition in aridlands? After fitting three different models to the data in aridec we found that there was not enough information to choose a unique model judging by their AICc values (Fig 2A). This limitation comes from the information contained in the original datasets which constrains our capacity to distinguish between models.

Simply put, we cannot force a model to reveal information that is not contained in the input data (Brun et al., 2001). As a workaround, we took a multi-model inference approach (Burnham and Anderson, 2002) that allowed us to incorporate the dynamics of all three models in our results by using AICc weights (Lukacs et al., 2010). In this way, our predictions of transit time in arid lands include the differences in litter chemistry and their effects on decomposition, instead of just considering the bulk of litter as a homogenous pool. This type of information

theoretical approach like model averaging is not novel, but is still underused in ecological studies (Grueber et al., 2011).

However, before fitting complex compartmental models, researchers should take into consideration the issue of collinearity. In a previous study, we found that most of the time series in the *aridec* database could only be fitted to simpler models with less than three parameters (Sarquis et al., 2022a). This was because the

information contained in those time series of litter decomposition was not sufficient to inform more complex models, for example: models with three distinct litter mass pools with transfer coefficients between them. This lack of information in the data caused collinearity between parameters, which in turn made it impossible to identify a single set of parameters for each model (Brun et al., 2001; Sierra et al., 2015). Some of these limitations probably come from the short number of sampling points in most decomposition studies (Sarquis et al., 2022a),

which lowers the degrees of freedom available and limits our capacity to model complex organic matter dynamics. The fact that complex models cannot be obtained from the data suggests that we should put more

attention into designing field experiments that can better inform about model structures that are more consistent with our current understanding of litter heterogeneity and transformations (Prescott and Vesterdal, 2021).

Our second question was: what are the potential ranges of the median transit times of C in litter for aridlands? This part of our study yielded some new insights into the biogeochemistry of arid environments. Median transit time from one- and two-pool decomposition models without interactions were similar and showed that half of the litter mass is lost after almost 2 years in the field (Fig. 3). However, results from the two-pool model with series structure were almost three times higher. This is explained by the mass transfer from the fast-

decomposing pool to the slow-decomposing pool, which slows down mass loss from litter. After model-averaging, we obtained intermediate values of median transit times of around three years (Fig. 3). Previously, estimations were made of mean transit time for litter of between 3.4 and 3.8 years for the same models as this study (Manzoni et al., 2012). However, their data did not come from an aridland. To our knowledge, our study is the first attempt to estimate litter transit time in arid environments.

The discrepancy between estimations from the two-pool series model and the other two models connects back to the issue of model parameter identifiability. Most decomposition studies carried out in aridlands last for only a year (Sarquis et al., 2022a). But our results show that decomposition of litter in arid environments can take on average six times longer until all litter mass exits the system. This means that most field decomposition studies are not capturing the entire dynamics of mass release through time. Most decomposition studies usually must

compromise between measurement resolution and study length. Usually, studies that describe fine-scale dynamics of chemical compounds in leaf litter do not last for the entire decomposition process. On the contrary, longer studies usually focus on broad-scale processes and represent litter as a homogenous pool. In turn, this has consequences for potential future research because the information that is not contained in data cannot be retrieved by modeling techniques (Brun et al., 2001). Similar to this study, Derrien and Amelung (2011)

concluded that future continuous isotope labelling studies should make more measurements in time and with a finer time resolution in order to make more reliable estimations of soil C fluxes and reservoirs from models. If we aim to incorporate field data into complex Earth system models, we need to take into consideration the study time length and resolution to capture both broad- and fine-scale mechanisms of decomposition. We acknowledge this might seem excessive given academic times go usually faster than litter decomposition in

aridlands. However, successful long-term litter decomposition projects exist and can be a potential solution to this issue (e.g.: LIDET; Gholz et al., 2000).

We asked as our third question: what are the relationships between median transit time and environmental variables? From the set of four variables that we used to explore these relationships, only mean annual temperature showed a moderate correlation with median transit time from average models (Fig. 4a). The

importance of temperature as a climatic driver of decomposition is well documented (Zhang and Wang, 2015), both through its positive effects on microbial activity (Sinsabaugh et al., 1991) and its increase of photochemical emissions (Day and Bliss, 2020). Moreover, the correlation with mean annual precipitation was weak (Fig. 4b). This was rather expected since it has been long known that precipitation fails to explain patterns of decomposition rates in aridlands (Austin, 2011).

As a final remark, we explored what can transit time teach us about the distribution of decomposition models. We calculated the quotient of the median transit time and the natural logarithm of two times the mean transit time from average models. Since the median of an exponential distribution equals ln2 times the mean, this ratio should equal one for models that are close to a single exponential distribution. But only 15 % of the models had values close to one (Fig. 5), which is indicative that for most cases models did not follow an exponential

distribution. The negative exponential model of decomposition has been the standard for litter and soil organic matter decomposition studies since at least five decades ago (Olson, 1963). This connects back to our first results where the one-pool exponential model was not chosen by our information theoretical approach (Fig. 2). Previous studies found similar results where the negative exponential one-pool model did not rank first for the entirety of the datasets considered (Adair et al., 2008; Cornwell and Weedon, 2014; Manzoni et al., 2012). One

alternative to exponential models has been a linear function relating mass loss and time, as it has performed statistically well in the past, especially in photodegradation experiments carried out in aridlands (Brandt et al., 2010). However, such linear functions lack any theoretical support as they imply that litter keeps losing mass even after all mass has decayed away in the long term. In contrast, the compartmental approach used here can account for chemical and physical transformations of litter as it decays and has strong theoretical support.

Future studies could take advantage of the compartmental modeling framework to test multiple model structures that would represent different mechanism of litter transformation and decay, having the one-pool model structure as a null model that can be contrasted against more complex structures suggested by the information content in the data.

**5 Conclusions**

Although our theoretical understanding of the litter decomposition process is based on the assumption that plant litter is chemically and physically heterogeneous, and undergoes multiple transformations, time series of litter decomposition studies contain only relatively little information on litter heterogeneity and its transformation rates. However, we have shown that a multi-model inference approach helps to reconcile theoretical understanding with information content in observed datasets of litter decomposition. In particular,

the combination of AIC model averaging applied to a metric that is independent of model structure, the transit time, provides an inference framework that is useful to understand decomposition dynamics. This framework could help us get a better insight into the chemical transformations of organic matter in litter and soil, and how soil organic matter responds to changes in the environment.

     We recognize some limitations for modelling these complex structures arise from field study designs that do not

capture the entire decomposition process. This limits the quantity and the quality of the information that can be extracted from empirical data. We recommend that future field decomposition studies incorporate in their designs some strategy to better capture the dynamics of different organic matter pools in litter. This could be done by either measuring the proportion of each compound through time, or by increasing sampling times and study length. The two latter can help gain a better fit and avoid collinearity when using an inverse-modelling

approach as in this study. We further encourage researchers to fit models other than the one-pool model, when possible.

**6 Appendices**

**Appendix A: entry name in the *aridec* database, study site, decimal coordinates and citation of the datasets included in this study.**

| Entry Name | Study Site | Coordinates | Citation |
|---|---|---|---|
| Austin2006a | Chubut, Argentina | Latitude: -45.7 Longitude: -70.3 | Austin et al. (2006) |
| Berenstecher2021 | Chubut, Argentina | Latitude: -45.7 Longitude: -70.3 | Berenstecher et al. (2021) |
| Brandt2007 | Colorado, USA | Latitude: 40.8 Longitude: -104.8 | Brandt et al. (2007) |
| Day2018 | Arizona, USA | Latitude: 33.5 Longitude: -111.8 | Day et al. (2018) |
| Giese2009 | Inner Mongolia, China | Latitude: 43.6 Longitude: 116.7 | Giese et al. (2009) |
| Huang2017 | Xinjiang, China | Latitude: 44.4 Longitude: 87.9 | Huang et al. (2017) |
| | Xinjiang, China | Latitude: 45.3 Longitude: 87.6 | |
| | Xinjiang, China | Latitude: 42.9 Longitude: 89.2 | |
| Li2016 | Inner Mongolia, China | Latitude: 43.0 Longitude: 120.7 | Li et al. (2016) |
| Manlay2004 | Kaolack, Senegal | Latitude: 13.8 Longitude: -15.7 | Manlay et al. (2004) |
| Qu2020a | Inner Mongolia, China | Latitude: 41.5 Longitude: 107.0 | Qu et al. (2020) |
| Santonja2017 | Provence-Alpes-Côte d'Azur, France | Latitude: 44.0 Longitude: 5.9 | Santonja et al. (2017) |
| Smith2018 | New Mexico, USA | Latitude: 32.5 Longitude: -106.8 | Smith and Throop (2018) |
| WangY2020 | Inner Mongolia, China | Latitude: 44.2 Longitude: 116.5 | Wang et al. (2020) |


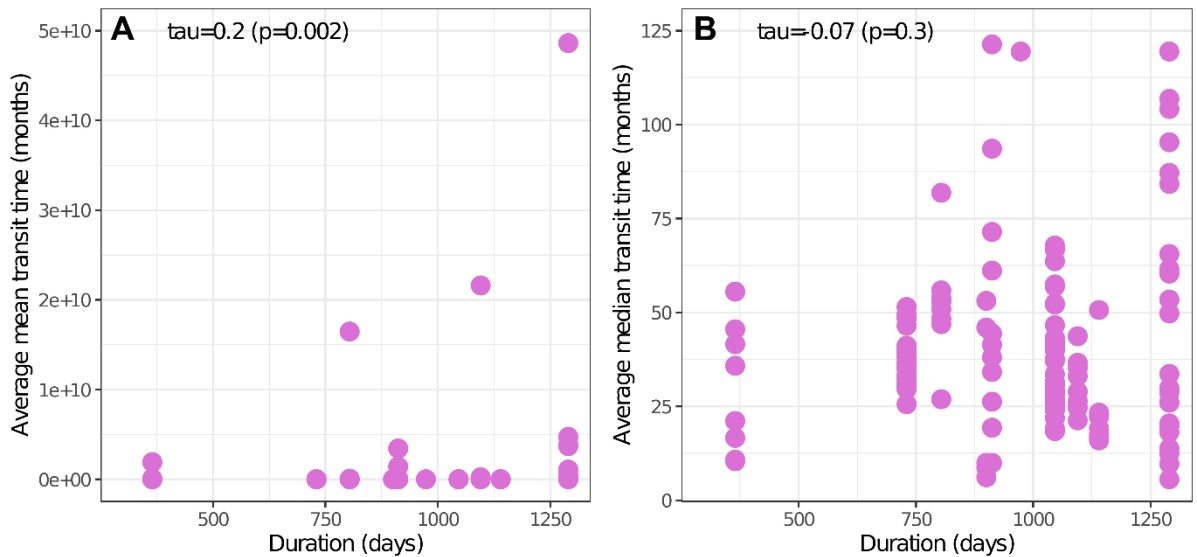

**Appendix B: non-parametric Kendall's rank correlation tests between study duration in days and avgMTT (A) and avgmTT (B), respectively.**

## 7 Code and data availability

The aridec database version 1.0.2 is archived and publicly available at https://doi.org/10.5281/zenodo.6600345 (Sarquis et al., 2022b). Result tables and code are stored at https://doi.org/10.5281/zenodo.7799585 (Sarquis and Sierra, 2023).

## 8 Author contribution

CAS supervised the study. CAS and AS conceptualized the study. AS curated the data and carried out the analysis.
AS wrote the original draft of the manuscript. CAS and AS revised and edited further versions of the manuscript.

## 9 Competing interests

The authors declare that they have no conflict of interest.

## 10 Acknowledgments

We thank two anonymous referees who made substantial contributions to the quality of this publication. We
thank Mina Azizi-Rad for early discussions on model fitting and selection. We also thank Ignacio Siebenhart for providing annual solar radiation data. We thank Amy Austin, Soledad Méndez and Ignacio Siebenhart for making revisions of early versions of this manuscript.

## 11 Financial support

Financial support for the project came from the University of Buenos Aires (grant no. UBACyT 2020), the Agencia
Nacional de Promoción Científica y Tecnológica (ANPCyT; projects PICT 2015-1231, PICT 2016-1780, and PICT 2019- 02645). Agustín Sarquis was funded by the University of Buenos Aires (UBACyT 2018; Res. No. 1245/18) and the German Academic Exchange Service (DAAD; Research Grants – Short-Term Grants program 2021, grant no. 57552337). Additional financial support was given by the Max Planck Society.

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
