# Peer review of "Information content in time series of litter decomposition studies and the transit time of litter in aridlands"

_EGUsphere, 2023_

## Author Response (AR1)

We would like to start by thanking the two anonymous referees that helped improve this manuscript with their comments. Aside from the changes suggested by the referees, we present in this new version a Table (Appendix A) with details on the study sites included in our manuscript. Answers to specific comments made by the referees are presented below.

**Anonymous Referee #1:**

**Comment 1:** What was the range of times (i.e., study length) to which these models were fit? I imagine most were short; did length of decomposition time affect results? By how much? Were mean/median transit times correlated at all with the length of study?
**Answer:** Study length ranged from 365 to 1290 days (mean of 974.1 and median of 1046 days). This is actually on the longer end of the whole range of study lengths in the aridec database (see figure 4 in Sarquis et al. 2022). Although we did not directly test this, we can infer that shorter studies were left out of our analysis when we made the median transit time cutoff to avoid abnormally large values. Usually, longer studies have more sampling dates, which in turn gives us more points to fit our models with a good performance (figures 5 and 6 in Sarquis et al. 2022). Information content in shorter studies might not be enough to fit our required models and thus their median transit time predictions were not reliable. Non-parametric Kendall's rank correlation between duration in days and the averaged mean transit time in this study was positive (tau=0.2, p=0.002). Instead, the same test for the average median transit time was not significantly different from 0 (p=0.3). From these tests we confirm that median transit time can be overestimated especially for longer studies, and this is why we chose to base our analysis on median transit time instead. We included these correlations in Methods (line 222), Results (line 262) and in Appendix A (line 380).

**Comment 2:** Are transit times used in other studies/systems? It would be interesting to know (e.g., in the discussion) if they correlate with the tested climate and litter quality variables in other or mesic systems.
**Answer:** Although transit times are used in other fields outside of biogeochemistry, e.g. hydrology, chemical engineering, or pharmacology (Sierra et al., 2017), its use in ecosystem ecology and biogeochemistry is somewhat limited. In biogeochemistry, the concept of residence time is widely adopted, but it is used in an inconsistent way. To our knowledge, there are only studies on transit times of carbon at the ecosystem level (Lu et al., 2018) or for the entire soil organic matter pool (Sierra et al., 2018). These studies have shown that transit times are usually in the order of one or two decades for warm and wet regions (tropical forest) and one or two centuries in cold and dry ecosystems such as tundra and arctic regions.

**Comment 3:** I couldn't find the metadata for the data and supplementary tables – it wasn't always clear to me what the headers meant or what the units were.
**Answer:** Thank you for this comment. We included a file "Metadata for CSV tables" in the Supplementary Material folder.

**Comment 4:** In the results and discussion (~L250 & 305), there is use and discussion of the quotient between the median and average transit times. In the results, I suggest explaining very clearly to the reader why this was done and why it is a useful index (I get it, but many may not). Generally, you don't want to make your readers work too hard - they may just give up and not get your point. I'd even be more explicit about what it says about the models and how they fit in the discussion – it won't be intuitive to all readers.

**Answer:** Thank you for this very good suggestion. We made some changes in the Results (line 275) and the Discussion (line 343) to make this easier to understand.

**Comment 5:** Did you look at other types of model forms, e.g., linear? That often works well for aboveground litter decomposition in aridlands.
**Answer:** We did not look at linear models. Our main focus was to compare between models with different compartments and interactions. We did a previous exploration of potential models with up to three pools and parallel, series or feedback interactions (see Sarquis et al., 2022). We tested for collinearity problems with those models and finally selected the models included in this current manuscript. We acknowledge that linear models have worked well in the past, especially in photodegradation experiments. We added a few words about this in the Discussion (line 351).

**Comment 6:** Finally, it may also be useful to see/discuss some common indicator of model fit for these three models (other than AICc) - did these models fit the data well? Or poorly?
**Answer:** For the 128 models we obtained root mean squared residuals. For the one-pool model this indicator ranged from 1.1 to 12.9, for the two-pool parallel model it ranged from 1.1 to 12.3, and for the two-pool series model it ranged from 0.3 to 6.6. This information was already available in the "Pre-averaging results.csv" table in the Supplementary material folder. However, thanks to your comment we added this information to Figure 2 and addressed this comment in line 242.

**Comment 7:** L30-31: it's not really clear from the abstract what this comment means – I get it after reading the manuscript, but if you could more clearly state what this means here it would help readers. Perhaps something along the lines of: "…our analysis suggests that current and historical litter decomposition studies often do not contain information on how litter quality changes over time or last long enough for litter to entirely decompose. This makes fitting accurate (or mechanistic?) models difficult (or impossible?)." Being a bit clearer will make the call for better experiments more compelling.
**Answer:** Thank you for your comment. We incorporated this modification in line 37.

**Comment 8:** L173: Reasonable decision, although the last few datapoints in that time series are (or can be) a bit odd (no need to respond to this).

**Comment 9:** L223-4: Suggest rephrasing (in brackets []): "Following this, we decided [to implement] a multi-model inference approach using model averaging, which left us with individual models (see table S3 for model variance and confidence intervals)."
**Answer:** Thank you for your comment. We incorporated this modification in line 246,

**Comment 10:** L230ish and Figure 4: Did the analysis include buried litter? i.e., were only "exposed" and "ambient" treatments used? This would impact the correlations with climate and solar radiation. Could you add this to the methods (either way)?
**Answer:** We only used ambient treatment data from all positions (exposed, covered and buried). We tested this as an option using only data from litter exposed to radiation, but correlation was still weak and either way it was not significant (r=0.2, p=0.6). We clarified the use of ambient data in Methods (line 227).

**Anonymous Referee #2:**

**Comment 11:** The paragraphs in the abstract where the different concepts and methods are addressed these are not so clearly described - Probably the text can be better understood if the concepts are connected. For example In L13-14, instead of writing "to this extent,

compartmental dynamical systems.." write "as an alternative, compartmental dynamical systems..".

**Answer:** Thank you for this suggestion. We incorporated it in line 13.

**Comment 12:** Further on L16 the term of "models with different structures" is mentioned but it is not clear what different structures mean "one-pool model" versus compartmental dynamical systems? Therefore, I would recommend that before introducing the concept of transit time to compare models with different structures, introduce the model structures themselves.

**Answer:** Thank you for this suggestion. We incorporated it in line 18.

**Comment 13:** L16 is it that transit time is the mean age or that transit time is the age.. and mean transit time is the mean age...? then "transit time" is defined associated to a compartmental system as if transit time could only be used on that type of model but it was already mentioned that it is an across-model metric. Is it that one single-pool system can also be considered a compartmental system? Otherwise write in L17 something like "…when they are released from a compartmental system or from the bulk litter in one-pool models."

**Answer:** Firstly, transit time is a random variable that describes the age of particles leaving a system. And from this random variable one can obtain a mean transit time which is the mean age of the particles in the output flux. We corrected this in the abstract (line 23). As for the second question, one-pool models can be considered as compartmental models with only one compartment. We hope the general changes in the abstract will now make this point clearer.

**Comment 14:** L20 what does parallel and series structures mean? It is not defined throughout the paper, what is their relation to the concept of compartmental dynamical systems?

**Answer:** Models with two or more parameters can have different structures, such as parallel if each compartment decomposes independently, or in series if there is mass transfer from one compartment to another. We added this information to the abstract in line 20.

**Comment 15:** L20 the first question was what model structures are more appropriate to represent decomposition from a publicly available database of decomposition studies. I do not think model averaging was a methodology that allowed to compare models and I think the first tool used is missing here, which is the AIC criteria that was actually used to first compare the three models. Therefore, my suggestion would be to add the used of the AIC comparison on in L20 and then to write "followed by model averaging as a multi-model inference approach".

**Answer:** Thank you for this suggestion. We incorporated it in line 25.

**Comment 16:** L21 median transit times of C were mentioned but I think it should be transit times of litter mass, right?

**Answer:** You are right, we clarified this in line 28.

**Comment 17:** L21 wouldn't it be better to define median transit time instead of mean transit time, because is the median that was used as the dependent variable?

**Answer:** We previously clarified in the abstract the definition of transit time as the age distribution of particles in the output flux, thanks to your suggestion. We do not mention mean transit time in this section anymore, hence the text is more cohesive in relation to the use of median transit time. Thank you again for making this suggestion.

**Comment 18:** In general, throughout the abstract, introduction and research questions the expression "model structures" is mentioned without clearly defining what the difference is between parallel and series structures and maybe also justifying why it is relevant to compare them and associate how the presented concepts of compartmental systems and transit time relate to the different structures.

**Answer:** Thank you for bringing up this observation. We understand it can be confusing, but it is nothing more than just a way to differentiate models from each other based on their parameters. Basically, the number of compartments, as well as their interactions determine model structure. To the extent of this study, the three different model structures equal the three candidate models. We elaborated on this in the abstract as you suggested in Comment 14. We additionally clarified this in the Introduction (line 65) and we had already explained this thoroughly in Methods.

**Comment 19:** In figure 1 add the meaning of the two vertical lines

**Answer:** Thank you for this comment. We added this information in line 159.

**Comment 20:** In figure one, the single-pool model shows the fast pool line, but instead of fast isn´t it the total pool that should be shown?

**Answer:** You are right, we corrected figure 1 as suggested.

**Comment 21:** L155 what is the TRY database?

**Answer:** TRY database was proposed with the explicit assignment to improve the availability and accessibility of plant trait data for ecology and earth system sciences. For more information on the history of its creation and its use see Kattge et al. (2020). We added a few lines about it (line 166).

**Comment 22:** L170 After introducing the mean and the median transit times it would be great to explain why the median should be used instead of the mean for assessing the models.

**Answer:** After calculating mean and median transit times we found that mean transit times were higher than median transit times in general. The average median transit time was 3.3 years while the average mean transit time was 110 million years. Needless to say, such time lengths exceed the possibilities of the litter decomposition process and are more in the realm of fossilization. This led us to decide that mean transit time was not a good indicator of decomposition for our dataset. Specifically, higher values of mean transit time corresponded to the more complex two-pool series model. This model has a mass transfer coefficient that adds a proportion of mass from the fast pool to the slow pool, and this clearly slows down the outflux from the litter system. We think that this issue and the already slow decomposition rates in these arid ecosystems altogether led to overestimations of this parameter. For this reason, we decided to work with median transit times instead. We clarified this issue in line 185.

**Comment 23:** L179 I think that would be the first method to answer the first research question

**Answer:** We agree with this statement. As you correctly suggested in Comment 15, we already clarified this in the Abstract. We also added this to the Introduction in line 108.

**Comment 24:** L190 What more information would be needed to be able to choose a model with this method?

**Answer:** This question is related to Information Theory. Data contains information, and this is always limited by study design and measurement procedures. Consequently, a particular data

set will only be able to inform a limited set of models. Simply put, we cannot force a model to reveal information that is not contained in the input data. One way in which we could obtain more information would be by increasing the number of sampling dates in the same period. However, in our case data comes from experiments that were already finished, so the information available to us is constrained. We included some lines on this subject in the Discussion (line 286).

**Comment 25:** L208 Does it make sense to use initial lignin content as both dependent variable (the second pool in the two-pool models) and independent variable (litter quality factor) and compare them??
**Answer:** The reviewer is correct, and it doesn't make sense to correlate the output of a model that uses initial lignin content as driving variable with this variable itself. Although we performed these correlations initially in an exploration phase before actually including initial lignin content in the model structure, we didn't remove this statement from the manuscript. The new version doesn't include lignin content as a variable used to correlate model output.

**Comment 26:** L220 I feel that the argument of values being very low sounds very relative and not enough to be used as a single argument to say that none of the models can be chosen. Is it possible ti cite someone that says how high the ΔAICc has to be to be reliable? Or isn´t it possible to perform a statistic to compare the three models in terms of their AICc??
**Answer:** As Burnham and Anderson (2002) state, there is not really a ΔAIC cutoff criterion to determine which model is the best. It is very common to find authors use a ΔAIC=2 cutoff criterion, but Burnham and Anderson state this is unbased and arbitrary. In our case, even if we wanted to use this criterion it would be impossible because there was never a single candidate model that differed from the other two with ΔAICc>2. Finally, as Anderson (2008) states: Information-theoretic approaches do not constitute a statistical "test" of any sort. There are no test statistics, assumed asymptotic sampling distributions, arbitrary a-levels, P-values, and arbitrary decision about "statistical significance". We added a few words regarding this in line 239.

**Comment 27:** L241 I would write "shows" instead of "suggests"
**Answer:** Thank you for this comment. We made this change in line 266.

**Comment 28:** L246 I would also display the significance levels of the correlations
**Answer:** Thank you for this suggestion. This is a big improvement for this figure.

**Comment 29:** L256 again I think here it would be a good opportunity to discuss more about what additional information was expected to decide between the models using the AICc methodology
**Answer:** Since you mentioned this in Comment 26, we further discussed this subject in line 321. Thank you for your comment.

**Comment 30:** L272-274 is a very good implication from this study.
**Answer:** Thank you for this comment. We also think this is a good implication from our study. This is why we mentioned it in the abstract, discussion and conclusions sections. We also talked about this in a previous publication (Sarquis et al., 2022).

**Comment 31:** L278 Is it half C or half litter mass?

**Answer:** This should be litter mass, thank your for your comment. We changed it accordingly (line 310).

**Comment 32:** L279-281 Can these decomposition times be validated with some examples of actual data using other techniques such as isotopes to track C for long terms?
**Answer:** Derrien & Amelung (2011) followed a similar approach to ours as they gathered information from the literature on C isotopes to calculate soil C mean residence times. They considered two models: a one-pool model and a two-pool model. Similarly to us, they found none of the two candidate models was the best choice in all cases, and they state: "To be able to select the best model for the mean residence time computation using continuous isotope labelling experiments, we recommend the recording of many data points, especially shortly after the isotopic change in order to capture any meaningful shift in the isotope pattern, and to carefully monitor the plant C input or the size of the reservoir". In short, it is possible to validate our results with isotope techniques, but the approach has limitations like the ones we encountered in our study. We added some of these ideas to the discussion (line 326).

**Comment 33:** L285 are there any studies that captured it ? (similar to my previous question)
**Answer:** We think that not many studies are able to capture the entire fine- and broad-scale dynamics of litter decomposition at the same time. Most decomposition studies usually must compromise between measurement resolution and study length. Usually, studies that describe fine-scale dynamics of chemical compounds in leaf litter do not last for the entire decomposition process. On the contrary, longer studies usually focus on broad-scale processes and represent litter as a homogenous pool. We are not aware of studies in arid lands that bridged this knowledge gap, but Prescott and Vesterdal (2021) reviewed this issue for forests. We added this to the Discussion in line 321.

**Comment 34:** L291 Are their results similar to the study results?
**Answer:** Referee No. 1 had a similar question in Comment 2. So far, not many studies have used transit time, and even fewer studies were made in relation to plant litter decomposition. We talk about one other study that did something similar to us with comparable results in lines 313.

**Comment 35:** In general, the study approach is very interesting because it is more integrative but as a non-modeler I found a bit of trouble understanding how these results can be more supported and/or validated or at least further discuss about the validation mechanisms with actual decomposition-times data.
**Answer:** We think this comment follows your above-mentioned comments regarding how our results fit next to previous studies. Firstly, we would like to remark that our study uses actual decomposition data from field experiments, as you mentioned. Secondly, models that integrate large datasets like in this case do not need to be validated with new empirical data (Oreskes, 1998). We do find, however that integrative modelling studies like this one can teach us plenty about general patterns and information voids. As we mentioned in the discussion, our study highlights the need to improve study designs of new field experiments to better capture mass loss dynamics of plant litter. We suggest that making longer experiments will allow to measure the entire decomposition curve in aridlands where decomposition can be especially slow. Additionally, making more point measurements during specific periods of decomposition will allow to capture fine-scaled processes that otherwise go unnoticed. These limitations apply to any type of study, whether it is organic mass loss of litter or carbon isotope tracing that we

consider. And of course, these limitations are then transferred to models. We thank you for your comment, since this is a very valuable discussion. We added some of these thoughts to line 329.

**Comment 36:** L8 Correspondence
**Answer:** Thank you for noticing this mistake. We changed it accordingly.

**Comment 37:** L19 and across the paper, why is aridlands as a single word? I would separate it.
**Answer:** Although not officially included in common English dictionaries, aridland is a common term in scientific publications (e.g.: Araujo et al., 2012; Austin, 2011; Ball et al., 2019; Collins et al., 2008; Reynolds et al., 2004; and the list goes on). We think that it does not make any difference either way. We appreciate your opinion, but we decided to keep it as it was.

**Comment 38:** L42 I think it should be written soil organic matter (SOM) formation and not SOC
**Answer:** Thank you for this suggestion. We changed it accordingly (line 50).

**Comment 39:** I would recommend for next time to add line numbers to all the lines and not only from 5 to 5 as it is easier to track the referred line when all of them are numbered.
**Answer:** We agree this would make it easier, but we did this following the Word template available in the journal's website (https://www.biogeosciences.net/submission.html# accessed on March 28th). For this reason, we decided to keep it in the original way.

**Comment 40:** Use indentation for the references.
**Answer:** We agree this makes it easier to read so we applied it. Thank you for this suggestion.

**Comment 41:** The names of the supplementary files are different than those stated on the list.
**Answer:** Thank you for this comment. We made this correction as suggested.

**Comment 42:** L190 the text of "the function" has a different size.
**Answer:** Thank you for noticing this mistake. We changed it accordingly (line 208).

**Comment 43:** L207 write the complete name of the correlation method to be more specific e.g. instead of "r" better "Pearson"
**Answer:** Thank you for this comment. We made this change in line 224.

**Comment 44:** L261 remove "C"
**Answer:** We thank you for this comment and made this change in the text.

**Bibliography**
Anderson, D. R.: Model Based Inference in the Life Sciences: A Primer on Evidence, Springer, New York., 2008.
Araujo, P. I., Yahdjian, L. and Austin, A. T.: Do soil organisms affect aboveground litter decomposition in the semiarid Patagonian steppe, Argentina?, Oecologia, 168(1), 221–230, doi:10.1007/s00442-011-2063-4, 2012.
Austin, A. T.: Has water limited our imagination for aridland biogeochemistry, Trends Ecol. Evol., 26(5), 229–235, doi:10.1016/j.tree.2011.02.003, 2011.
Ball, B. A., Christman, M. P. and Hall, S. J.: Nutrient dynamics during photodegradation of plant litter in the Sonoran Desert, J. Arid Environ., 160(January 2018), 1–10, doi:10.1016/j.jaridenv.2018.09.004, 2019.
Burnham, K. P. and Anderson, D. R.: Model Selection and Multimodel Inference, edited

by K. P. Burnham and D. R. Anderson, Springer New York, New York, NY., 2002.

Collins, S. L., Sinsabaugh, R. L., Crenshaw, C., Green, L., Porras-Alfaro, A., Stursova, M. and Zeglin, L. H.: Pulse dynamics and microbial processes in aridland ecosystems, J. Ecol., 96(3), 413–420, doi:10.1111/j.1365-2745.2008.01362.x, 2008.

Derrien, D. and Amelung, W.: Computing the mean residence time of soil carbon fractions using stable isotopes: impacts of the model framework, Eur. J. Soil Sci., 62(2), 237–252, doi:10.1111/j.1365-2389.2010.01333.x, 2011.

Kattge, J., Bönisch, G., Díaz, S., Lavorel, S., Prentice, I. C., Leadley, P., Tautenhahn, S., Werner, G. D. A., Aakala, T., Abedi, M., Acosta, A. T. R., Adamidis, G. C., Adamson, K., Aiba, M., Albert, C. H., Alcántara, J. M., Alcázar C, C., Aleixo, I., Ali, H., Amiaud, B., Ammer, C., Amoroso, M. M., Anand, M., Anderson, C., Anten, N., Antos, J., Apgaua, D. M. G., Ashman, T., Asmara, D. H., Asner, G. P., Aspinwall, M., Atkin, O., Aubin, I., Baastrup-Spohr, L., Bahalkeh, K., Bahn, M., Baker, T., Baker, W. J., Bakker, J. P., Baldocchi, D., Baltzer, J., Banerjee, A., Baranger, A., Barlow, J., Barneche, D. R., Baruch, Z., Bastianelli, D., Battles, J., Bauerle, W., Bauters, M., Bazzato, E., Beckmann, M., Beeckman, H., Beierkuhnlein, C., Bekker, R., Belfry, G., Belluau, M., Beloiu, M., Benavides, R., Benomar, L., Berdugo-Lattke, M. L., Berenguer, E., Bergamin, R., Bergmann, J., Bergmann Carlucci, M., Berner, L., Bernhardt-Römermann, M., Bigler, C., Bjorkman, A. D., Blackman, C., Blanco, C., Blonder, B., Blumenthal, D., Bocanegra-González, K. T., Boeckx, P., Bohlman, S., Böhning-Gaese, K., Boisvert-Marsh, L., Bond, W., Bond-Lamberty, B., Boom, A., Boonman, C. C. F., Bordin, K., Boughton, E. H., Boukili, V., Bowman, D. M. J. S., Bravo, S., Brendel, M. R., Broadley, M. R., Brown, K. A., Bruelheide, H., Brumnich, F., Bruun, H. H., Bruy, D., Buchanan, S. W., Bucher, S. F., Buchmann, N., Buitenwerf, R., Bunker, D. E., et al.: TRY plant trait database – enhanced coverage and open access, Glob. Chang. Biol., 26(1), 119–188, doi:10.1111/gcb.14904, 2020.

Lu, X., Wang, Y.-P., Luo, Y. and Jiang, L.: Ecosystem carbon transit versus turnover times in response to climate warming and rising atmospheric CO2 concentration, Biogeosciences, 15(21), 6559–6572, doi:10.5194/bg-15-6559-2018, 2018.

Oreskes, N.: Evaluation (not validation) of quantitative models., Environ. Health Perspect., 106(suppl 6), 1453–1460, doi:10.1289/ehp.98106s61453, 1998.

Prescott, C. E. and Vesterdal, L.: Decomposition and transformations along the continuum from litter to soil organic matter in forest soils, For. Ecol. Manage., 498(July), 119522, doi:10.1016/j.foreco.2021.119522, 2021.

Reynolds, J. F., Kemp, P. R., Ogle, K. and Fernández, R. J.: Modifying the 'pulse–reserve' paradigm for deserts of North America: precipitation pulses, soil water, and plant responses, Oecologia, 141(2), 194–210, doi:10.1007/s00442-004-1524-4, 2004.

Sarquis, A., Siebenhart, I. A., Austin, A. T. and Sierra, C. A.: Aridec : an open database of litter mass loss from aridlands worldwide with recommendations on suitable model applications, Earth Syst. Sci. Data, 14(7), 3471–3488, doi:10.5194/essd-14-3471-2022, 2022.

Sierra, C. A., Müller, M., Metzler, H., Manzoni, S. and Trumbore, S. E.: The muddle of ages, turnover, transit, and residence times in the carbon cycle, Glob. Chang. Biol., 23(5), 1763–1773, doi:10.1111/gcb.13556, 2017.

Sierra, C. A., Hoyt, A. M., He, Y. and Trumbore, S. E.: Soil Organic Matter Persistence as a Stochastic Process: Age and Transit Time Distributions of Carbon in Soils, Global Biogeochem. Cycles, 32(10), 1574–1588, doi:10.1029/2018GB005950, 2018.